# Mechanisms underlying sharpening of visual response dynamics with familiarity

Sukbin Lim[1,2]*

[1]Neural Science, NYU Shanghai, Shanghai, China; [2]NYU-ECNU Institute of Brain and Cognitive Science, NYU Shanghai, Shanghai, China

**Abstract** Experience-dependent modifications of synaptic connections are thought to change patterns of network activities and stimulus tuning with learning. However, only a few studies explored how synaptic plasticity shapes the response dynamics of cortical circuits. Here, we investigated the mechanism underlying sharpening of both stimulus selectivity and response dynamics with familiarity observed in monkey inferotemporal cortex. Broadening the distribution of activities and stronger oscillations in the response dynamics after learning provide evidence for synaptic plasticity in recurrent connections modifying the strength of positive feedback. Its interplay with slow negative feedback via firing rate adaptation is critical in sharpening response dynamics. Analysis of changes in temporal patterns also enables us to disentangle recurrent and feedforward synaptic plasticity and provides a measure for the strengths of recurrent synaptic plasticity. Overall, this work highlights the importance of analyzing changes in dynamics as well as network patterns to further reveal the mechanisms of visual learning.
DOI: https://doi.org/10.7554/eLife.44098.001

## Introduction

Experience-dependent changes in neural responses have been suggested to underlie the more efficient and rapid processing of stimuli with learning. Human and monkeys have been reported to process familiar stimuli with shorter response times and with less effort (*Greene and Rayner, 2001*; *Logothetis et al., 1995*; *Mruczek and Sheinberg, 2007*). The possible neural correlate for such behavior enhancement is the sharpening of stimulus selectivity that is achieved by broadening the distribution of activities as the stimulus becomes familiar (*Freedman et al., 2006*; *Kobatake et al., 1998*; *Lim et al., 2015*; *McKee et al., 2013*; *Woloszyn and Sheinberg, 2012*). Also, temporal sharpening of neural responses with experience has been observed, which can increase the resolution of discriminating stimuli in time with learning (*Meyer et al., 2014*; *Recanzone et al., 1992*).

Modifications of synaptic connections have been thought to be one of the basic mechanisms for learning. A repeated encounter of a stimulus would elicit a particular activity pattern in the network, which in turn modifies synaptic connections depending on pre- and post-synaptic activities. Such modifications of synaptic connections lead to changes in neural responses that can be a substrate to differentiate learned and unlearned stimuli. The previous modeling works investigated the relationship between synaptic plasticity and changes in network activity to find a synaptic plasticity rule that can account for sharpening of stimulus selectivity observed with learning (*Dayan and Abbott, 2005*; *Gerstner and Kistler, 2002*). However, whether such rules can also explain temporal changes in neural responses is in question.

In this work, we investigate the mechanism underlying changes of response dynamics with learning. To this end, we consider neural activities recorded in inferior temporal cortex (ITC) known to be important for visual object recognition (*Miyashita, 1993*; *Tanaka, 1996*). In ITC, changes in the response properties with learning have been reported in several experiments (*Freedman et al., 2006*; *Li et al., 1993*; *Lim et al., 2015*; *Logothetis et al., 1995*; *McKee et al., 2013*; *Woloszyn and*

*For correspondence:
sukbin.lim@nyu.edu

Competing interests: The author declares that no competing interests exist.

*Sheinberg, 2012*; *Xiang and Brown, 1998*). The average over different visual stimuli of time-averaged responses decreases with familiarity, while the distribution of responses across visual stimuli broadens with learning. The dynamics of visual responses were also found to change with familiarity – in particular, rapid successive presentation of familiar images, but not novel images, elicits strong periodic responses (*Meyer et al., 2014*).

Previously, we investigated synaptic plasticity in recurrently connected circuits that reproduces changes in the distribution of time-averaged visual responses observed experimentally (*Lim et al., 2015*). As the distribution of time-averaged visual responses of a single cell to multiple stimuli can be a surrogate for a spatial pattern of the network to one stimulus in a homogeneous network, the previous work mainly focused on how recurrent synaptic plasticity shapes the network pattern and stimulus tuning. Here, we extend our previous framework to understand mechanisms underlying changes of temporal patterns with learning. First, we demonstrate that the synaptic plasticity rule inferred from the time-averaged responses is not sufficient to reproduce changes in response dynamics. Next, we show that the interaction between synaptic plasticity and negative feedback mechanisms is critical for generating stronger oscillation after learning. Using a mean-field analysis, we identify the conditions on synaptic plasticity and negative feedback to reproduce changes in response dynamics consistently observed in different experimental settings. Finally, we validate these conditions through network simulations and infer the post-synaptic dependence of synaptic plasticity from the experimental data.

## Results

### Effects of visual learning on response dynamics

In this section, we summarize the effects of visual experience on response dynamics obtained from three different laboratories comparing the visual response to novel (unlearned) and familiar (learned) stimuli in the monkey ITC (*Lim et al., 2015*; *McKee et al., 2013*; *Meyer et al., 2014*; *Woloszyn and Sheinberg, 2012*). Two experiments measured visual responses to the presentation of one stimulus, one in a passive viewing task (*Lim et al., 2015*; *Woloszyn and Sheinberg, 2012*) and the other in a dimming-detection task (*Freedman et al., 2006*; *Lim et al., 2015*; *McKee et al., 2013*). The duration of the stimulus presentation and number of stimuli were different in the two tasks: shorter duration of stimulus presentation and a larger set of stimuli in the passive viewing task in comparison to the dimming-detection task (Materials and methods). In both cases, the average response to familiar stimuli was lower than that to novel stimuli with a rapid decrease of the response around 150 ms after the stimulus onset in putative excitatory neurons (*Figure 1A*; Figure 5A,B for the dimming-detection task). On the other hand, the response to the most preferred stimulus was found to increase for familiar stimuli with broadening of the distribution of time-averaged activities (*Figure 1B,C*).

In both the mean and maximal responses to familiar stimuli, a rebound of activity was observed around 230 ms after the stimulus onset (*Figure 1A,B*). This is distinctive from responses to novel stimuli showing slow decay after the transient rise. We further quantified the magnitude of the rebound before and after learning by measuring the slope of changes in the activities at each rank of stimuli (*Figure 1D*). It showed that the higher-rank familiar stimuli exhibit the stronger rebound in putative excitatory neurons. In contrast, there is only a weak dependence between the rank of stimuli and the magnitude of rebound activity in inhibitory neurons (*Figure 1—figure supplement 1*).

The emergence of oscillatory responses after learning was also observed in different experimental settings. In the dimming detection task with longer stimulus presentation, the average response showed damped oscillation for familiar stimuli (Figure 5B; *Freedman et al., 2006*; *McKee et al., 2013*). In another experiment where either two novel stimuli or two familiar stimuli were presented rapidly in sequence, the peak response for the second familiar stimulus is as strong as the one for the first stimulus, while the response to the novel stimulus is suppressed at the second peak (*Meyer et al., 2014*). Thus, rapid successive presentation of familiar images, but not novel images, elicits strong periodic responses. Note that although all three experiments suggest stronger oscillation after learning, its strength may vary depending on a sampling of neurons and stimuli as only excitatory neurons with their most preferred stimuli exhibit strong oscillation after learning (*Figure 1D*).

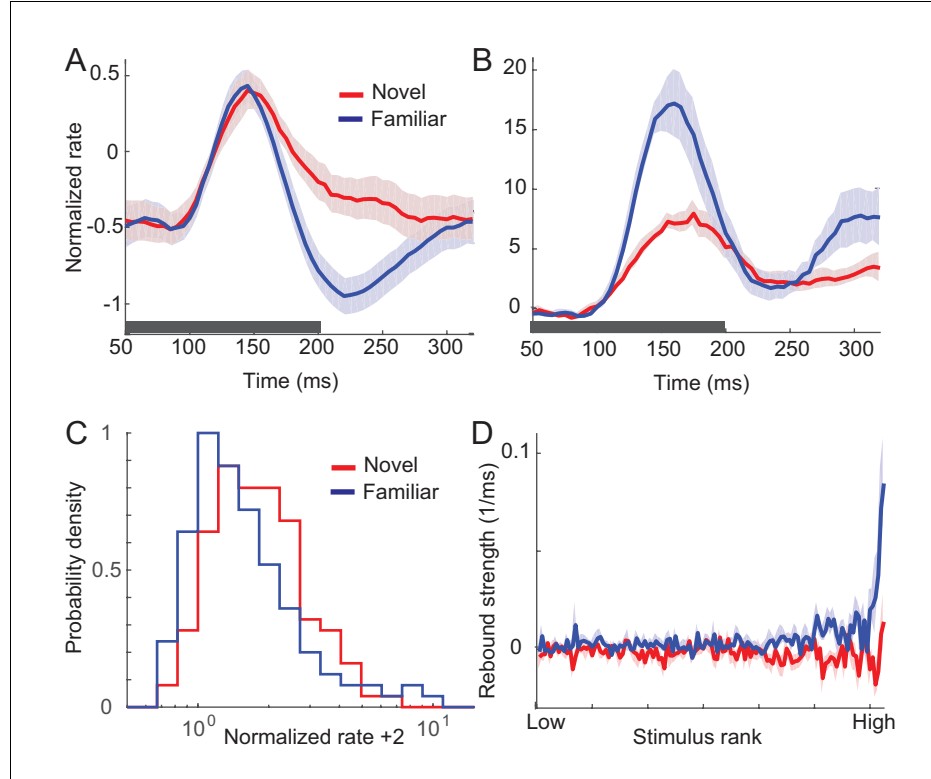

**Figure 1.** Changes in response dynamics of putative excitatory neurons with learning in a passive viewing task (*Lim et al., 2015*; *Woloszyn and Sheinberg, 2012*). (**A, B**) Average and maximal response to familiar (blue) and novel (red) stimuli. For each excitatory neuron, we normalized firing rates by the mean and standard deviation of time-averaged activities over novel stimuli during stimulus presentation (80 ms-200 ms after the stimulus onset) and took the average over stimuli (**A**) and the response with the highest time-averaged activity (**B**). Solid curves are activities averaged over neurons, and shaded regions represent mean ± s.e.m of activities averaged over individual neurons. The gray horizontal bar represents the visual stimulation period starting at 0 ms. (**C**) Distribution of time-averaged activities during stimulus presentation. For each neuron, according to time-averaged activities, a stimulus was rank-ordered among familiar and novel stimuli, respectively. At each rank of the stimuli, we averaged the normalized response over neurons, and obtained the distributions of activities over different ranks of stimuli. To avoid negative values in the x-axis on a logarithmic scale, we added two to normalized rates. (**D**) Rebound strength of damped oscillation. At each rank of stimuli, the rebound strength was quantified by the slope of changes of activities between 230 ms and 320 ms after the stimulus onset.

DOI: https://doi.org/10.7554/eLife.44098.002

The following source data and figure supplement are available for figure 1:

**Source code 1.** Data for *Figure 1D*.
DOI: https://doi.org/10.7554/eLife.44098.019

**Figure supplement 1.** Changes in response dynamics of putative inhibitory neurons with learning in a passive viewing task (*Woloszyn and Sheinberg, 2012*).
DOI: https://doi.org/10.7554/eLife.44098.003

In sum, the prominent effects of visual learning on responses of excitatory neurons are (i) reduction in average response, (ii) increase in maximum response, and (iii) stronger oscillations after learning. In the following, we show how such changes guide us to reveal a mechanism underlying visual learning that sharpens stimulus selectivity and temporal resolution of stimuli. Note that we focus on excitatory neurons only assuming that the dynamics of inhibitory neurons follow that of mean excitatory neurons, and do not contribute qualitative changes of response dynamics after learning. Such a simplification is based on the experimental observation that input changes and the magnitude of rebound activity depend weakly on the post-synaptic firing rates in inhibitory neurons (See Discussion for further justification).

## Recurrent synaptic plasticity alone cannot reproduce the response dynamics

Activity-dependent modifications of synaptic connections can be one of the key elements to explain changes in network patterns and response dynamics with learning. Previously, we introduced a procedure to infer synaptic plasticity rules from experimental data so that networks implementing the derived learning rules can quantitatively reproduce changes in the distribution of time-averaged visual responses observed experimentally (*Lim et al., 2015*). We now extend this framework and explore whether synaptic plasticity alone would be sufficient to explain stronger oscillatory responses after learning.

To investigate the effect of learning on response dynamics, we considered a firing rate model with a plasticity rule that modifies the strength of recurrent synapses as a function of the firing rates of pre- and postsynaptic neurons. Activities of neurons are described by their firing rates $r_i$ for $i = 1,\ldots, N$, where $N$ denotes the number of neurons in the network. Their dynamics are described by the following equations

$$\tau_r \frac{dr_i}{dt} = -r_i + \Phi\left(\sum W_{ij}^R r_j + \sum W_{ij}^F I_j^X\right) \tag{1}$$

where $\Phi$ is a static transfer function (*f-I* curve), and the total input current is the sum of the recurrent input $\sum W_{ij}^R r_j$ and the feedforward input $\sum W_{ij}^F I_j^X$. $W_{ij}^k$ denotes the strength of synaptic connection from neuron $j$ to neuron $i$ with $k = R$ or $F$ representing recurrent and feedforward connections, respectively. The superscript $X$ denotes an external input, and $I_i^X$ is the external input to neuron $i$ before learning with $W_{ij}^F = \delta_{ij}$.

We assumed that the recurrent synapses are plastic, changing their strengths according to $W_{ij}^R \rightarrow W_{ij}^R + \Delta W_{ij}^R$, which depends on the activities of both pre- and postsynaptic neurons during the stimulus presentation. We further assumed that the learning rule is a separable function of pre- and postsynaptic activity as

$$\Delta W_{ij}^R = \frac{1}{N} f_R(\xi_i) g_R(\xi_j) \tag{2}$$

where $f$ and $g$ are post- and pre-synaptic dependence of the learning rules, respectively, and $\xi_i$ is the activity of neuron $i$ averaged during the stimulus presentation before learning.

Previously, we found that synaptic plasticity in recurrent excitatory connections is sufficient to reproduce changes in the distribution of time-averaged visual responses observed experimentally (*Lim et al., 2015*). Hebbian-type synaptic plasticity with a potentiation in high firing rates leads to an increase of the maximal response of excitatory neurons, while overall depression leads to a decrease of the average network response of both excitatory and inhibitory neurons (*Figure 2A*). With such synaptic plasticity derived from the time-averaged activities, response dynamics in *Equation (1)* shows similar changes to the time-averaged responses (*Figure 2B,C*). However, the temporal profile is similar before and after learning and does not show oscillations after learning. Thus, synaptic plasticity alone is not sufficient for reproducing changes in response dynamics observed experimentally, which will be shown analytically in the next section.

## Interactions between recurrent synaptic plasticity and slow negative feedback

Another key ingredient to explain changes in response dynamics with learning can be slow negative feedback. In a dynamical system, resonance-like behavior emerges from the interaction between strong positive feedback and relatively slow negative feedback. Thus, enhanced resonance behavior after learning observed experimentally may suggest that changes in synaptic connections strengthen positive feedback in the circuit and affect the response dynamics by interacting with a slow negative feedback mechanism. Also, the reduced response to successive stimulus presentation of novel stimuli (*Meyer et al., 2014*) can be caused by the slow recovery from negative feedback.

For generating a damped oscillatory response after learning, we found that specific negative feedback such as firing rate adaptation is required (*Figure 3*). Similar to previous works investigating the effect of adaptation on the network activity in a mean-field approach (*Fuhrmann et al., 2002*; *Laing and Chow, 2002*; *Tabak et al., 2006*; *Treves, 1993*; *van Vreeswijk and Hansel, 2001*), we

considered a linear mechanism for adaptation where the adaptation current is a low-pass filtered firing rate represented by the variable $a_i$ with time constant $\tau_a$ and strength $k$. Then the dynamics of network activity is described by the following equations:

$$\tau_r \frac{dr_i}{dt} = -r_i + \Phi\left(\sum W_{ij}^R r_j - k a_i + \sum W_{ij}^F I_j^X\right)$$
$$\tau_a \frac{da_i}{dt} = -a_i + r_i$$

(3)

Intuitively, interactions between recurrent synaptic plasticity and adaptation-like negative feedback in *Equations (2) and (3)* can reproduce two effects of visual learning, increase in maximal response and stronger oscillatory response after learning. Hebbian-type synaptic plasticity in recurrent connections provides strong potentiation in the connections among high firing rate neurons, and thus, generates a cell assembly with stronger positive feedback after learning (*Figure 3A*). This leads to not only an increase in the response of this cell assembly but also the emergence of oscillation under the interplay with slow adaptation currents. The strength of oscillation in the rest of the population may depend on the synaptic strengths from these high firing rate neurons.

To show this analytically, we investigated mean-field dynamics that summarize network activity with fewer variables (Materials and methods). To facilitate the analysis, we made two assumptions, linear dynamics with transfer function $\Phi(x) = x$, and homogeneous connectivity before learning that reflects no correlation between novel stimuli and network structure. Under these assumptions, the dynamics before learning is described by average activity and adaptation, $\bar{r} = \frac{1}{N}\sum_i r_i$ and $\bar{a} = \frac{1}{N}\sum_i a_i$.

After learning, with synaptic plasticity in recurrent connections following *Equation (2)*, recurrent connections become correlated with activity pattern they learned. Increased correlation between the learned pattern and network structure can be captured by additional variables $m$ and $n$, defined as $m = \frac{1}{N}\sum_i g_R(\xi_i)r_i$ and $n = \frac{1}{N}\sum_i g_R(\xi_i)a_i$, which is a variation of the pattern overlap $\frac{1}{N}\sum_i \xi_i r_i$ utilized previously to describe changes in dynamics with learning (*Tsodyks and Feigel'man, 1988*).

The variables $m$ and $n$ can approximately represent the activities and adaptation of high firing rate neurons as the activities and adaptation of high firing rate neurons contribute more to $m$ and $n$ variables with monotonically increasing pre-synaptic dependence $g_R(\xi_i)$ (*Figure 3A*). Thus, potentiation of recurrent inputs in high firing rate neurons provides strong positive feedback in $m$, while slow adaptation mechanisms represented by $n$ variables provide negative feedback. As the variables $m$ and $n$ are only present in the dynamics after learning, qualitative changes of the response dynamics in the network should be mainly led by their dynamics with strong potentiation in high rate neurons (*Figure 3B*). Such strong potentiation and generation of damped oscillation in high rate neurons are consistent with the observation that the rebound is strongest in those neurons (*Figure 1D*).

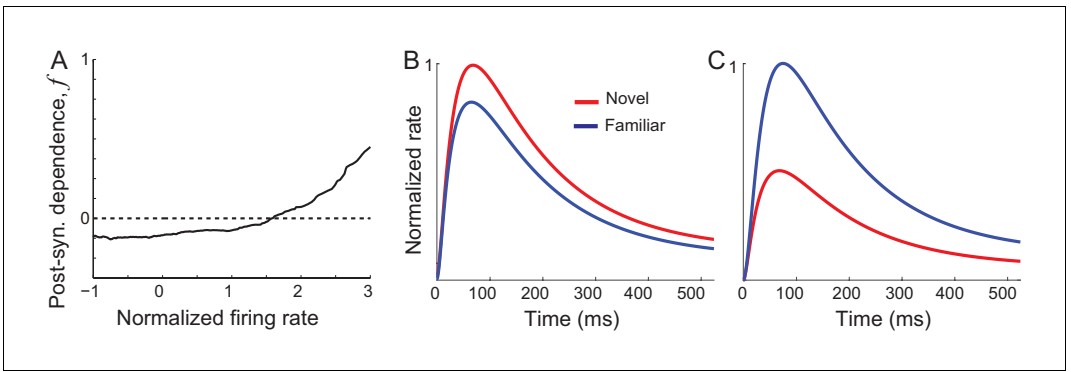

**Figure 2.** Networks with synaptic plasticity in recurrent connections without slow negative feedback. (**A**) Example post-synaptic dependence of recurrent synaptic plasticity inferred from changes of time-averaged responses. Dependence of synaptic plasticity on the post-synaptic rate, $f$ in *Equation (2)*, shows depression for low rates and potentiation at high rates. (**B, C**) Average (**B**) and maximal (**C**) response before (red) and after (blue) learning for the network with synaptic plasticity only in recurrent connections.
DOI: https://doi.org/10.7554/eLife.44098.004

The recurrent input from high rate neurons can lead to a damped oscillatory response in the rest of the population (**Figure 3A**). The mean-field analysis shows that the strength of the damped oscillatory response is proportional to the strength of postsynaptic synaptic plasticity $f_R(\xi_i)$ in the case of linear dynamics. If $f_R$ for neuron $i$ is positive (negative) corresponding to potentiation (depression) in recurrent inputs, an oscillation in neuron $i$ would be in phase (out of phase) with that of high rate neurons. Previously, we proposed Hebbian-type but overall depression-dominant synaptic plasticity in recurrent connections to minimally account for the decrease in time-averaged responses (**Lim et al., 2015**). However, this would lead to out of phase oscillation in the mean and maximum response, inconsistent with the data (**Figure 1A,B**). Instead, overall potentiation in recurrent inputs with $\bar{f}_R > 0$ is required to generate in-phase oscillation in the mean and maximum response in linear dynamics (**Figure 3—figure supplement 1**).

## Additional synaptic plasticity for reduction in average response

We showed that recurrent synaptic plasticity could account for the emergence of damped oscillation and sharpening neural activities by increasing the maximal response after learning. Furthermore, synchronous oscillations in the mean and maximum response observed experimentally suggest overall potentiation in recurrent inputs. However, potentiation-dominant synaptic plasticity in recurrent connections would increase overall synaptic input and cannot reproduce a decrease in average activities with learning (**Figure 3—figure supplement 1A,B**). The same holds for recurrent synaptic plasticity with or without the assumption of the constant sum normalization which imposes a constraint on the pre-synaptic dependence (**Figure 3—figure supplement 2**).

Instead, reduction in average response requires changes in external inputs or other recurrent inputs such as suppression in other excitatory inputs or enhanced inhibition. Enhanced recurrent inhibition can result from an increase in inhibitory activities after learning or potentiated inhibitory connections onto excitatory neurons. The former is inconsistent with the experimental observations showing a reduction in inhibitory firing rates across different stimuli (**Figure 1—figure supplement 1**). Also, potentiated inhibition with learning is less likely to account for a decrease of average

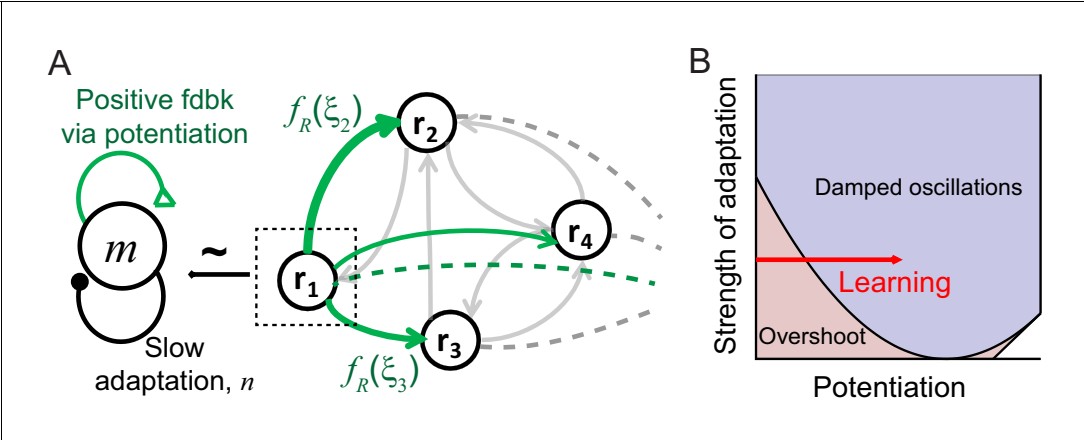

**Figure 3.** Mechanism of generating damped oscillations after learning. (**A**) Schematics of the dynamics after learning. The overlap variable $m$ is similar to activities of high rate neurons represented as $r_1$, and $n$ represents adaptation in $m$. These high rate neurons drive a damped oscillation in the remaining population whose strength is proportional to the post-synaptic dependence of recurrent synaptic plasticity $f_R(\xi)$. (**B**) Interactions between potentiation of recurrent inputs and a slow adaptation mechanism. The strength of potentiation is proportional to $\overline{fg_R}$ (**Equation 4**) which is 0 before learning. The separatrix dividing overshoot and damped oscillations is shown as a parabola defined by $\overline{fg_R}$, the strength of adaptation $k$ and time constants $\tau_R$ and $\tau_A$.

DOI: https://doi.org/10.7554/eLife.44098.005

The following figure supplements are available for figure 3:

**Figure supplement 1.** Networks with synaptic plasticity only in recurrent connections with slow negative feedback.
DOI: https://doi.org/10.7554/eLife.44098.006

**Figure supplement 2.** Networks with synaptic plasticity only in recurrent connections without the constraint of the sum normalization of synaptic weights achieved by $\sum g_k(\xi) = 0$.
DOI: https://doi.org/10.7554/eLife.44098.007

excitatory activities - a temporal profile of inhibitory activities after learning shows a decrease of activity almost to the baseline in the late phase of the stimulus presentation (200–250 ms after the stimulus onset). This suggests that the effect of potentiated inhibition in the late phase is weaker than in the early phase while reduction of excitatory activities was observed in the late phase (*Figure 1A,B*).

Another possibility is a depression in recurrent excitation through different types of synapses such as potentiation in fast AMPA-like currents and depression in slow NMDA-like currents. Depression in slow excitatory currents can lead to a decrease in excitatory activities in the late phase. However, different regulation of AMPA and NMDA currents is inconsistent with the experimental observations showing maintenance of a constant NMDA-to-AMPA ratio under the changes of AMPA receptors induced chemically or by an STDP protocol (*Watt et al., 2000*; *Watt et al., 2004*).

Instead of additional changes of the recurrent synaptic inputs, we considered changes in external inputs with feedforward synaptic plasticity $\Delta W_{ij}^F = f_F(\xi_i)g_F(\xi_j)$. Together with overall potentiation in the recurrent connections, dominant depression in the feedforward connections with $\bar{f}_F < 0$ can reproduce the reduction of average responses over the stimuli with learning. In *Figure 4*, an example network with Hebbian learning rule in recurrent connections, uniform depression in the feedforward connection, and spike adaptation mechanisms was shown to reproduce the effects of visual learning qualitatively. With learning, the average response decreases in particular in the late phase (*Figure 4A*), but maximal firing rates increased and oscillation becomes prominent especially in high rates (*Figure 4B,C*). Also, in the successive presentation of two stimuli, the average response shows stronger oscillation after learning (*Figure 4D*), while the rank of individual neuronal activities changes when a new stimulus arrives (*Figure 4E,F*).

Note that the mean field dynamics was derived under the assumption of linear dynamics. With synaptic or neuronal nonlinearity, some conditions identified through our mean field dynamics can be mitigated such as less dominant potentiation in recurrent inputs with learning (*Figure 6—figure supplement 1*). However, a network simulation with example nonlinearity still shows that the core principles on the synaptic plasticity rule remain the same as strong potentiation in recurrent connections in high rate neurons, and average depression in feedforward inputs.

## Network simulation and comparison with data

In this section, we validate that network models implementing the conditions identified through mean-field equations indeed reproduce the experimental observation and allow us to infer the post-synaptic dependence of synaptic plasticity. To illustrate this, we considered electrophysiological data obtained in a passive viewing task and dimming-detection task (*Lim et al., 2015*; *McKee et al., 2013*; *Woloszyn and Sheinberg, 2012*). In the dimming detection task, responses to fewer stimuli were measured, and we considered the response averaged over neurons and stimuli, which was fitted using mean-field dynamics (*Figure 5*). The external inputs and parameters of the $\bar{r}$ and $\bar{a}$ dynamics before learning were chosen to generate no oscillations (*Figure 5A*; *Figure 5—figure supplement 1*). Potentiation in high firing rate neurons, average potentiation of recurrent inputs and depression in feedforward inputs were found to mimic response to familiar stimuli (*Figure 5B*).

This mean-field dynamics reproduces prominent features of response dynamics before and after learning, showing damped oscillation and a decrease in average response to familiar stimuli after its peak (*Figure 5C*). Furthermore, we simulated the mean response to novel and familiar stimuli for a successive presentation of stimuli (*Figure 5D*). When novel stimuli are repeatedly shown, the peak response to the second stimuli is smaller than the response to the first, due to a slow recovery from the adaptation current. In contrast, for the serial presentation of familiar stimuli, the response to the first stimulus decays quickly and the response to the second stimulus is less affected by the adaptation current. Thus, the overall response becomes more oscillatory compared to the one for novel stimuli.

In the experimental data obtained during the passive viewing task, the duration of stimulus presentation was shorter, but the distribution of response dynamics before and after learning could be obtained (*Figure 1C*; Materials and methods). As in the dimming detection task, the external inputs were obtained from the responses to novel stimuli. By comparing the response dynamics at each rank of the novel and familiar stimuli, we derived the post-synaptic dependence of synaptic plasticity in recurrent and feedforward connections. Note that the synaptic plasticity was inferred from

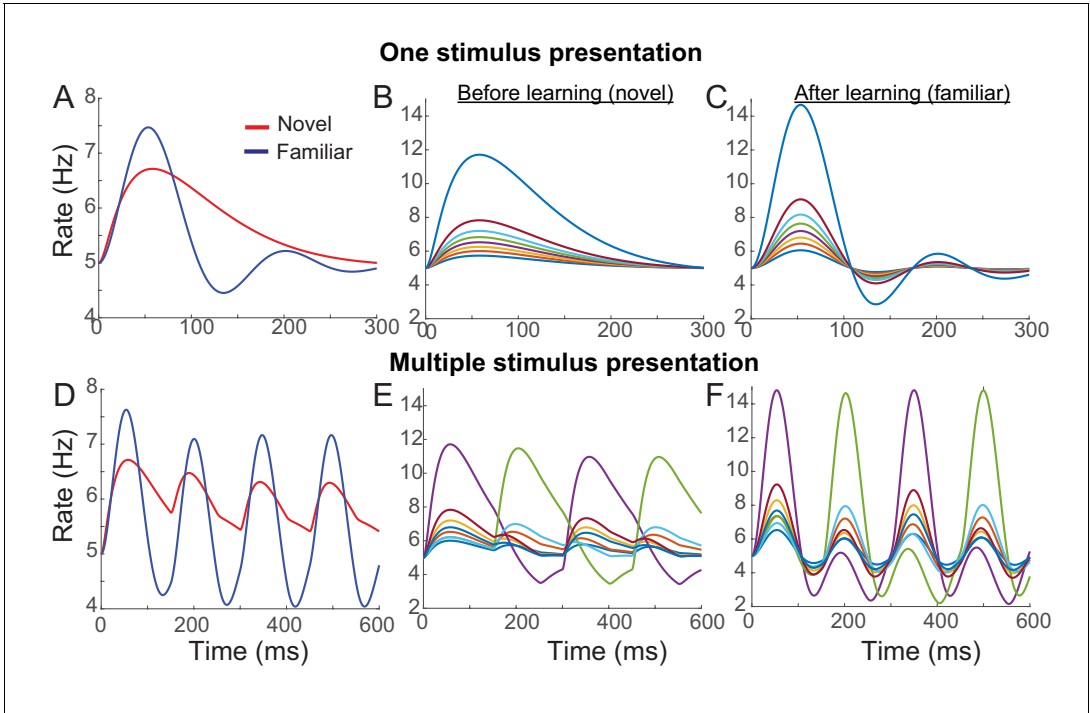

**Figure 4.** Example network reproducing the effects of visual learning in one stimulus presentation (**A–C**) and successive presentation of two stimuli (**D–F**). The network implements potentiation in the recurrent connections through Hebbian synaptic plasticity, depression in the feedforward connections through uniform scaling down of the external inputs, and spike-adaptation mechanisms. Mean responses reproduce the effects of visual learning predicted in the mean-field dynamics, showing average reduction and stronger oscillations (**A, D**). Representative individual activities before (**B,E**) and after (**C,F**) learning show that activities in neurons with high firing rates increase with strong oscillation after learning (**E,F**), but the rank of stimuli is shuffled with the arrival of a new stimulus.

DOI: https://doi.org/10.7554/eLife.44098.008

The following source code is available for figure 4:

**Source code 1.** MATLAB code for *Figure 4*.
DOI: https://doi.org/10.7554/eLife.44098.020

normalized firing rates averaged over neurons under the assumption that the dependence of synaptic plasticity rules on normalized firing rates is the same across different neurons (See Discussion for justification).

Consistent with the fitting of the mean-field dynamics to the data obtained in a dimming detection task, the average post-synaptic dependence of synaptic plasticity leads to potentiation in recurrent inputs and depression in feedforward inputs (*Figure 6A*). Furthermore, the post-synaptic dependence in recurrent connections is an increasing function of the rank of stimuli, or equivalently, the post-synaptic activities. It is notable that such a tendency is similar to the dependence of the rebound magnitude to familiar stimuli on the rank of stimuli observed experimentally (*Figure 1D*). Network models implementing the derived synaptic plasticity reproduce the reduction of average activities (*Figure 6B*) and rebound in the late phase of stimulus presentation in both average and maximal responses, although the maximal response after the initial rise is less well fitted (*Figure 6C*).

We also checked whether the key conditions for synaptic plasticity change with example nonlinear input-output transfer function derived from the time-averaged response to novel stimuli (*Figure 6—figure supplement 1A*). The derived post-synaptic dependence in both recurrent and feedforward connections is similar to that obtained under linear dynamics with more balance between depression and potentiation in recurrent synaptic plasticity (*Figure 6—figure supplement 1B*). Although the rebound in the average activities is less well fitted compared to that with linear dynamics, the network simulations agree with the data qualitatively (*Figure 6—figure supplement 1C,D*).

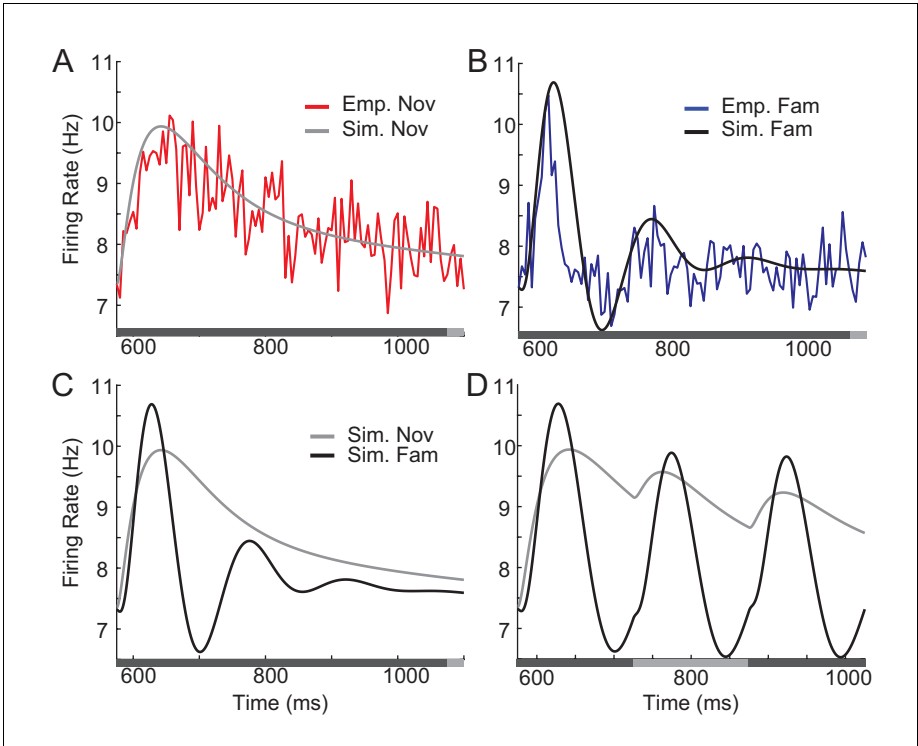

**Figure 5.** Comparison between network simulation and data obtained in a dimming-detection task. (**A, B**) Fitting response dynamics (red in A and blue in B) using mean-field equations (gray in A and black in B) for novel (**A**) and familiar (**B**) stimuli. (**C,D**) Simulation for one stimulus presentation (**C**) and successive presentation of stimuli (**D**). The gray horizontal bar represents the visual stimulation period starting at 500 ms and x-axis is truncated to show activities from their onsets. In A-C, the stimulus was presented for a duration that was a sum of a fixed duration (650 ms shown in the dark gray) and a random duration (shown in the light gray). In D, different gray bars represent different stimuli shown alternatively for a duration of 150 ms.

DOI: https://doi.org/10.7554/eLife.44098.009

The following source data and figure supplements are available for figure 5:

**Source code 1.** Data for *Figure 5*.
DOI: https://doi.org/10.7554/eLife.44098.021

**Source code 2.** MATLAB code for *Figure 5*.
DOI: https://doi.org/10.7554/eLife.44098.022

**Figure supplement 1.** Constraints on $w_R$ and $k$ to reproduce responses to novel stimuli (shaded area).
DOI: https://doi.org/10.7554/eLife.44098.010

**Figure supplement 2.** Parameter search for the strengths of potentiation and adaptation (A,B) and average post-synaptic dependence of recurrent and feedforward connections, $\bar{f}_R$, and $\bar{f}_F$ (C).
DOI: https://doi.org/10.7554/eLife.44098.011

**Figure supplement 3.** Sensitivity of fitting to changes in the recurrent connectivity strength before learning, $w_R$.
DOI: https://doi.org/10.7554/eLife.44098.012

## Alternative negative feedback mechanisms

Our mean field analysis and model fit to the data suggest firing rate adaptation mechanisms as a good candidate for slow negative feedback to explain the familiarity effect on the dynamics. Here, we explored whether two alternative negative feedback mechanisms such as delayed global inhibition or short-term depression can replace adaptation. Delayed global inhibition may arise due to local inhibition with slow NMDA- or GABA$_B$-like currents in inhibitory feedback pathways, or inhibitory feedback from other areas. For instance, prefrontal cortex shows a familiarity effect with a long

latency around 100 ms but with opposite sign (*Rainer and Miller, 2000*; *Xiang and Brown, 2004*), and thus, the top-down signals from this area can serve as slow negative feedback.

We considered a model of global inhibition so that all excitatory neurons receive the same slow inhibition whose strength is proportional to the average activity of excitatory neurons (Materials and methods). Under the assumption of linearity, we could derive the mean field equations similar to that with adaptation mechanisms with variables $r$, $a$ and $m$ but without variable $n$ that mainly represents the negative feedback in high firing rate neurons (*Equation (6)*). Without negative feedback, $m$ cannot generate damped oscillations after learning in both high rate neurons and the overall population. This suggests that slow negative feedback private to individual neurons or sub-populations is required to generate qualitative changes in dynamics as interacting with synaptic plasticity.

Short-term depression in synaptic connections has also been suggested as a mechanism for negative feedback and generating oscillations in cortical circuits (*Laing and Chow, 2002*; *Loebel and Tsodyks, 2002*; *Tabak et al., 2006*; *Wang, 2010*). To see whether short-term depression can reproduce the damped oscillatory response after learning, we considered a phenomenological model mimicking the effect of depletion of a neurotransmitter such that when the pre-synaptic firing is high, the synaptic input from such neuron becomes weak due to the lack of resources (Materials and methods; *Tsodyks and Markram, 1997*). Under the assumption that the recurrent connection is weak before learning, and the damped oscillation in the network is led by that in the high rate neurons, we searched for a parameter set of the strength and timescale of short-term plasticity that provides the best fit to the experimental data. However, the network simulation with the best-fitted parameters cannot generate a strong rebound, unlike the adaptation mechanisms (*Figure 6—figure supplement 2*). Thus, a simple phenomenological model of short-term plasticity cannot explain the qualitative changes in response dynamics observed experimentally.

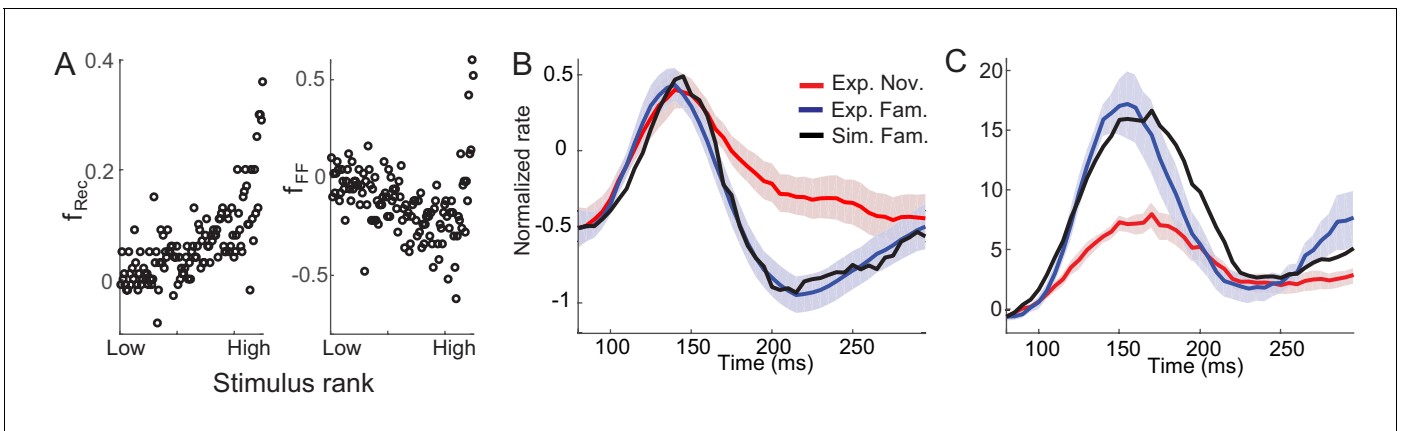

**Figure 6.** Post-synaptic dependence of synaptic plasticity in recurrent (left) and feedforward (right) connections derived from a passive viewing task (A) and comparison between the data and network simulations for average (B) and maximal (C) responses. The external inputs were chosen so that the response to novel stimuli is the same in the experiment and simulation (red in B,C). With the derived post-synaptic dependence in the recurrent and feedforward connections (A), the response to familiar stimuli was simulated (black in B,C).

DOI: https://doi.org/10.7554/eLife.44098.013

The following source data and figure supplements are available for figure 6:

**Source code 1.** MATLAB code for *Figure 6*.
DOI: https://doi.org/10.7554/eLife.44098.023

**Figure supplement 1.** Example nonlinearity in dynamics and synaptic plasticity inferred under nonlinearity.
DOI: https://doi.org/10.7554/eLife.44098.014

**Figure supplement 2.** Short-term depression cannot reproduce a damped oscillation after learning.
DOI: https://doi.org/10.7554/eLife.44098.015

**Figure supplement 3.** Schematics of synaptic plasticity rules in different connections.
DOI: https://doi.org/10.7554/eLife.44098.016

# Discussion

In this work, we provided a mechanistic understanding of how interactions between synaptic plasticity and a negative feedback mechanism implementing firing rate adaptation shape response dynamics with learning. The emergence of damped oscillations after learning requires strong positive feedback through potentiation in recurrent connections particularly among neurons with high firing rates. Such recurrent synaptic plasticity broadens the distribution of activities, while depression in feedforward inputs decreases average firing rates. Synaptic plasticity, therefore, enables the sparse and efficient representation of the learned stimuli. Furthermore, the strength of rebound of damped oscillation observed after learning can be a novel, graded measure for recurrent synaptic plasticity. On the other hand, adaptation-like mechanisms are critical for enhanced oscillatory responses after learning, and strongly suppresses the neural activities for the learned stimuli in particular in the late phase of the stimulus presentation. As such temporal sharpening prepares neurons to respond to the subsequent stimulus, our work suggests that the adaptation mechanisms together with synaptic plasticity may play an important role in the rapid processing of the learned stimuli.

Here, we extended our previous work inferring recurrent synaptic plasticity rules from time-averaged data in a static model of a cortical network to time-course data and a dynamic model with additional spike adaptation mechanisms and feedforward synaptic plasticity (*Lim et al., 2015*). Analyzing time-course data allows disentangling contributions of synaptic plasticity in different connections. However, similar to the previous work, only post-synaptic dependence of the synaptic plasticity rules can be inferred from single cell recordings under the assumption that the learning rules are a separable function of pre- and postsynaptic rates. Also, fitting the time course poses a limitation such that synaptic plasticity rules needed to be inferred from the data averaged over neurons due to noise. On the other hand, time-averaged data allows to infer recurrent synaptic plasticity in different neurons, which reveals a strong correlation between neural activity and the threshold separating depression and potentiation, but no correlation when the post-synaptic activity is normalized (*Lim et al., 2015*). Inspired by this observation, we inferred synaptic plasticity rules from normalized firing rates under the assumption that synaptic plasticity rules are the same across different neurons when inputs and rates are normalized (*Figures 1* and *6*). Although a direct test of this assumption is not feasible, the relatively small variance of rebound strengths over neurons may support this assumption on the recurrent synaptic plasticity as the dependence of rebound strengths on the rank of stimuli alternatively represents learning rules in recurrent connections (*Figure 6C*). Furthermore, if the learning rule inferred from the time-averaged response is the combination of recurrent and feedforward synaptic plasticity, the same learning rules of this mixture and recurrent connections across different neurons would justify the assumption on the feedforward plasticity (*Figure 6—figure supplement 3*).

Our work provides a reconciling perspective between two prominent classes of synaptic plasticity models suggested for familiarity detection and associative memory in ITC. Depression in the feedforward connections required to lower average response after learning reasserts the role of feedforward synaptic plasticity suggested for familiarity detection (*Bogacz and Brown, 2003*; *Norman and O'Reilly, 2003*; *Sohal and Hasselmo, 2000*). On the other hand, most theoretical works implementing synaptic plasticity in recurrent connections have focused on associative memory and the emergence of attractors with learning (*Amit and Brunel, 1997*; *Pereira and Brunel, 2018*; *Sohal and Hasselmo, 2000*). Unlike most of the previous works focusing on one-type of synaptic plasticity, our analysis proposed that both recurrent and feedforward synaptic plasticity are required to reproduce changes in spatial and temporal patterns underlying familiarity detection. A recent study investigated the memory capacity for associative memory under recurrent synaptic plasticity whose form was derived from neural activities related to familiarity detection (*Pereira and Brunel, 2018*). Similarly, it can be further investigated how the feedforward and recurrent synaptic plasticity rules derived from the data for familiarity detection contribute to other types of memory, and how a memory capacity changes dynamically during the stimulus presentation with slow spike adaptation mechanisms.

As a substrate for slow negative feedback, firing rate adaptation mechanisms have been suggested to be critical in generating network oscillations and synchrony (*Ermentrout et al., 2001*; *Fuhrmann et al., 2002*; *La Camera et al., 2004*; *Laing and Chow, 2002*; *Tabak et al., 2006*; *van Vreeswijk and Hansel, 2001*; *Wang, 2010*), and in optimal information transmission under a

particular form of synaptic plasticity (*Hennequin et al., 2010*). The effect of synaptic plasticity on enhancing synchrony in the recurrent synaptic circuits also has been explored theoretically (*Gilson et al., 2010*; *Karbowski and Ermentrout, 2002*; *Morrison et al., 2007*). Consistent with these previous works, our work suggests that the interplay between synaptic plasticity and adaptation with the time constant consistent with that of cellular adaptation mechanisms (*Benda and Herz, 2003*) generate synchronous damped oscillations after learning. Note that our analysis based on the data obtained from single cell physiology is limited to firing rate synchrony, and how spike-time correlation between neurons changes with visual learning needs to be further explored. Also, our work emphasizes the role of adaptation in different types of recognition memory. Previously, the adaptation mechanisms in the temporal cortex have been suggested to encode the recency of stimuli, which is typically measured by suppression of the response to the repetition of a stimulus (*Meyer and Rust, 2018*; *Miller et al., 1991*; *Vogels, 2016*; *Xiang and Brown, 1998*). As the time scale of repetition suppression lasts up to seconds, it may require the adaptation mechanisms on the much longer time scale (*Sanchez-Vives et al., 2000*). Thus, adaptation on various time scales (*La Camera et al., 2006*; *Pozzorini et al., 2013*) may be required for different types of recognition memory.

In our work, we assumed that inhibition minimally contributes to shaping response dynamics with learning for the following reasons. First, no dependence of input changes on post-synaptic firing rates in inhibitory neurons observed experimentally suggests that changes in inhibitory activities with learning can reflect the reduction of average excitatory activities and thereafter, excitatory inputs to inhibitory neurons without synaptic plasticity in the excitatory (*E*)-to-inhibitory (*I*) connections (*Lim et al., 2015*). On the other hand, anti-Hebbian synaptic plasticity in the *I*-to-*E* connections can have similar effects as Hebbian-synaptic plasticity in the *E*-to-*E* connections. Alternatively, overall potentiation in the *I*-to-*E* connections can provide stronger negative feedback or can replace the role of feedforward synaptic plasticity. However, as the dynamics of inhibitory neurons show strong suppression almost to the baseline in the late phase of the stimulus presentation after learning (*Figure 1—figure supplement 1*), neither anti-Hebbian synaptic plasticity nor potentiation can account for an increase of maximal response of excitatory neurons in the early phase and overall reduction in activities in the late phase (*Figure 1*). Thus, we assumed that changes in the inhibitory pathway are less likely to induce oscillation or suppression in the excitatory neurons. It is notable that the interaction between synaptic plasticity in both recurrent excitatory and inhibitory connections was suggested to reproduce increased transient response with learning (*Moldakarimov et al., 2006*). Although the homeostatic inhibitory plasticity proposed in this work cannot reproduce damped oscillatory response observed in ITC, we cannot rule out the role of inhibitory synaptic plasticity that can be complementary to the mechanisms proposed in our work.

The enhanced oscillation for familiar stimuli investigated here was around 5 Hz, which is in the range of theta oscillations. Such a low-frequency oscillation has been discussed in visual

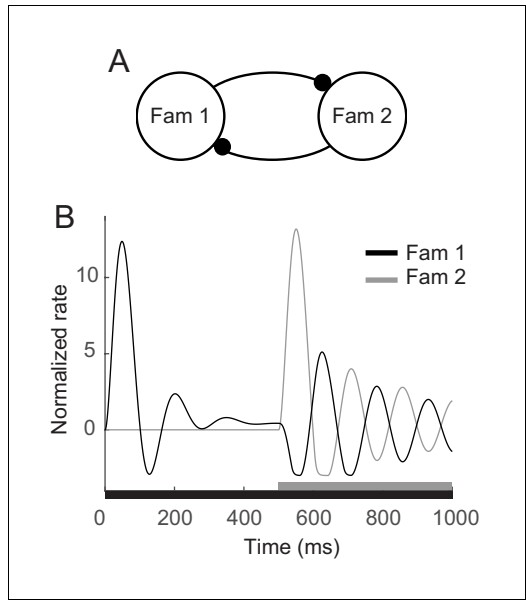

**Figure 7.** Enhanced oscillation through competitive interactions between different stimuli. (**A**) Two mutually inhibitory populations selectively responding to stimuli 1 and 2, respectively. The dynamics of each population follows the dynamics of *m* in the mean-field description for a single familiar stimulus in *Figures 3A,5*. (**B**) Time course of visual responses of two populations with different stimulus onsets (black and gray bars below). Stimulus 2 was present 500 ms after the onset of stimulus 1, and the population two was assumed to be silent before the arrival of stimulus 2. After the onset of stimulus 2, visual response selective to stimulus one was transiently suppressed and showed stronger oscillation compared to that under the single stimulus presentation.

DOI: https://doi.org/10.7554/eLife.44098.017

search to characterize overt exploration or sampling behaviors such as saccadic or microsaccadic eye movements (*Otero-Millan et al., 2008*; *Buzsaki, 2011*) and to underlie covert shift of attention that samples different stimuli rhythmically (*Dugué et al., 2015*; *Fiebelkorn et al., 2013*; *Landau and Fries, 2012*). In line with these studies, *Rollenhagen and Olson (2005)* observed that low-frequency oscillation became stronger when another stimulus was present together. Competitive interactions between populations representing different stimuli were suggested to generate oscillation with fatigue mechanism (*Moldakarimov et al., 2005*; *Rollenhagen and Olson, 2005*). Based on the adaptation mechanisms proposed in the current work, competition between two different familiar stimuli can generate stronger oscillation at a similar frequency. In the mean-field dynamics with two mutually inhibitory populations each of which mimics the maximum response to a single familiar stimulus, stronger oscillation but with the similar frequency with that for the single stimulus presentation was reproduced in the presentation of two stimuli (*Figure 7*). This may indicate low-frequency damped oscillators for a single familiar stimulus can be a building block for a rhythmic sampling of multiple stimuli and covert attentional shift through competitive interactions.

Overall, our work resonates with perspectives emphasizing the importance of dynamics in understanding cognitive functions (*Bargmann and Marder, 2013*; *Kopell et al., 2014*). As an extension of our previous work that inferred the synaptic plasticity rules from changes in spatial patterns, additional analysis of response dynamics revealed the role of slow adaptation currents in shaping response dynamics. Different contributions to activity changes of recurrent and feedforward synaptic plasticity suggested in our work can be further utilized to examine how each synaptic plasticity engages during the progress of learning. Also, although we suggested a local circuit model for visual learning, interactions with other areas might also be important – for instance, the interactions between ITC and perirhinal cortex may form positive feedback given the adjacency of these two areas and similar familiarity effects observed experimentally (*Xiang and Brown, 1998*). On the other hand, prefrontal cortex showing opposite effects of familiarity with a long latency may provide slow negative feedback (*Xiang and Brown, 2004*). To dissect the interaction between multiple regions, one can analyze time course data investigating latencies and qualitative changes in dynamics in these areas such as the emergence of oscillatory response after learning.

## Materials and methods

### Mean-field dynamics

To derive the mean field dynamics from *Equation (3)*, we assumed linear dynamics with $\Phi(x) = x$ and uniform recurrent connectivity before learning $W_{ij}^R = w_R/N$. Note that uniform connection can be replaced by random connection, which is analogous to the state where the network connectivity is stabilized after learning of a large number of uncorrelated activity patterns, but not correlated with the stimulus of interest (*Lim et al., 2015*). We also assumed $\sum g_k(\xi_j) = 0$ so that the sum of synaptic weights over the presynaptic neurons is preserved with learning. The external input to neuron $i$ before learning is defined as $I_i^X$ with $W_{ij}^F = \delta_{ij}$.

Before learning, the mean-field dynamics can be obtained by taking an average over neurons, which yields a two-dimensional system of differential equations in terms of $\bar{r} = \frac{1}{N}\sum_i r_i$ and $\bar{a} = \frac{1}{N}\sum_i a_i$. After learning, with $W_{ij}^k \rightarrow W_{ij}^k + \frac{1}{N}f_k(\xi_i)g_k(\xi_j)$, *Equation (3)* becomes

$$\tau_r \frac{dr_i}{dt} = -r_i + w_r \frac{1}{N}\sum_j r_j + f_R(\xi_i)\frac{1}{N}\sum_j g_R(\xi_j)r_j - ka_i + I_i^X + f_F(\xi_i)\frac{1}{N}\sum g_F(\xi_j)I_j^X$$
$$\tau_a \frac{da_i}{dt} = -a_i + r_i$$

The mean-field dynamics is four-dimensional with additional variables $m = \frac{1}{N}\sum_i g_R(\xi_i)r_i$ and $n = \frac{1}{N}\sum_i g_R(\xi_i)a_i$. The dynamics of $m$ and $n$ can be obtained by multiplying $g_R(\xi_j)$ to *Equation (3)* and taking the average over neurons as

$$\tau_r \frac{d\frac{1}{N}\sum_j g_R(\xi_j)r_{j_i}}{dt} = -\frac{1}{N}\sum_j g_R(\xi_j)r_j + w_r\frac{1}{N}\sum_j g_R(\xi_j)\frac{1}{N}\sum_j r_j + \frac{1}{N}\sum_j f_R(\xi_j)g_R(\xi_j)m$$
$$-k\frac{1}{N}\sum_j g_R(\xi_j)a_j + k\frac{1}{N}\sum_j g_R(\xi_j)I_j^X + \frac{1}{N}\sum_j f_F(\xi_j)g_R(\xi_j)I_{FX}$$

$$\tau_a \frac{d\frac{1}{N}\sum_j g_R(\xi_j)a_j}{dt} = -\frac{1}{N}\sum_j g_R(\xi_j)a_j + \sum_j g_R(\xi_j)r_j$$

With $\sum g_k(\xi_j) = 0$, the second term in the first equation disappears (note that without $\sum g_k(\xi_j) = 0$, this term remains and provides feedback from $\bar{r}$ to the $m$ dynamics as in *Figure 3—figure supplement 2*). Then, the mean-field dynamics after learning is given as

$$\tau_R \frac{d\bar{r}}{dt} = -\bar{r} + w_R\bar{r} + \bar{f}_R m - k\bar{a} + \bar{I}_X + \bar{f}_F I_{FX}$$
$$\tau_A \frac{d\bar{a}}{dt} = -\bar{a} + \bar{r}$$
$$\tau_R \frac{dm}{dt} = -m + \overline{fg}_R m - kn + I_{MX} + \overline{fg}_F I_{FX} \qquad (4)$$
$$\tau_A \frac{dn}{dt} = -n + m$$

with

$$\bar{r} = \frac{1}{N}\sum_i r_i, \qquad \bar{a} = \frac{1}{N}\sum_i a_i$$
$$m = \frac{1}{N}\sum_i g_R(\xi_i)r_i, \quad n = \frac{1}{N}\sum_i g_R(\xi_i)a_i. \qquad (5)$$

In *Equation (4)*, $\bar{f}_{R,F} = \frac{1}{N}\sum_i f_{R,F}(\xi_i)$ is the average post-synaptic dependence of recurrent and feedforward synaptic plasticity, $\overline{fg}_k = \frac{1}{N}\sum_i f_k(\xi_i)g_R(\xi_i)$ is the average of the product of post- and pre-synaptic dependence $f$ and $g$. $I$ represent the external inputs where $\bar{I}_X = \frac{1}{N}\sum I_i^X, I_{FX} = \frac{1}{N}\sum g_F(\xi_i)I_i^X, I_{MX} = \frac{1}{N}\sum g_R(\xi_i)I_i^X$.

To describe visual responses under the successive presentation of stimuli (*Meyer et al., 2014*), we consider learning of two stimuli. Changes of synaptic connections after two stimuli become $W_{ij}^k \to W_{ij}^k + \Delta W_{ij}^{k,1} + \Delta W_{ij}^{k,2}$ where superscripts 1 and 2 represent the indices of the stimuli. With the same synaptic plasticity rule as in *Equation (2)*, $\bar{f}_{R,F}$ and $\overline{fg}_{R,F}$ for different stimuli are the same, and the external input is the sum of the inputs $I_i^X = I_i^{X,1} + I_i^{X,2}$. For simplicity, we assume that the interaction of learning two stimuli is minimal such that two stimuli are uncorrelated as $\sum f_k(\xi_j^{l_1})g_k(\xi_j^{l_2}) = 0$ and $\sum I_j^{X,l_1} g_k(\xi_j^{l_2}) = 0$ for $l_1, l_2 = 1,2$ but $l_1 \neq l_2$. Then, by defining the overlap variables as $m = \frac{1}{N}\sum(g_R(\xi_j^1) + g_R(\xi_j^2))r_j$, $n = \frac{1}{N}\sum(g_R(\xi_j^1) + g_R(\xi_j^2))a_j$, and inputs as $I_{FX} = \frac{1}{N}\sum(g_F(\xi_i^1) + g_F(\xi_i^2))I_i^X$ and $I_{MX} = \frac{1}{N}\sum(g_R(\xi_i^1) + g_R(\xi_i^2))I_i^X$, the dynamics after learning two stimuli is the same as for that stimulus given in *Equation (4)*.

## Constraints on parameters in the mean-field dynamics

In this section, we describe the conditions on parameters in the mean-field dynamics *Equation (4)* to reproduce changes of response dynamics with learning qualitatively. Changes in response dynamics showing stronger oscillation after learning imposes a condition on the $m$ and $n$ dynamics in *Equation (4)*, and thus the constraints on the strength of potentiation, $\overline{fg}_R$, parameters for adaptation, $k$ and $\tau_A$, and time constant $\tau_R$ (*Figure 3B*). Also, response dynamics to novel stimuli such as no damped oscillation before learning and reduced response in the successive presentation of novel stimuli leads to constraints on the dynamics of $\bar{r}$ and $\bar{a}$ before learning, thus, $k$, $\tau_A$, $\tau_R$, and connectivity strength before learning $w_R$ (*Figure 5—figure supplement 1*).

Under the linear assumption, the dynamics is characterized by the eigenvalues of the system, and the eigenvalues of the $m$ and $n$ dynamics are given as $(-1+\overline{fg_R})/\tau_R - 1/\tau_A \pm \sqrt{\{(-1+\overline{fg_R})/\tau_R + 1/\tau_A\}^2 - 4k/(\tau_R\tau_A)}$. The transition to overdamped oscillation occurs when the eigenvalue becomes a complex number, that is, $\{(-1+\overline{fg_R})/\tau_R + 1/\tau_A\}^2 - 4k/(\tau_R\tau_A)$ changes its sign. This provides a separatrix (red to blue region in *Figure 3B*). Also, the stability requiring a negative real part of eigenvalues imposes two other conditions, $\overline{fg_R} < 1 + \tau_R/\tau_A$, and $(-1+\overline{fg_R})/\tau_R - 1/\tau_A + \sqrt{\{(-1+\overline{fg_R})/\tau_R + 1/\tau_A\}^2 - 4k/(\tau_R\tau_A)} < 0$, that is, $\overline{fg_R} - k - 1 < 0$ (two lines on the right side in *Figure 3B*).

Similarly, the eigenvalues of linear dynamics of $\bar{r}$ and $\bar{a}$ before learning are given as $(-1+w_R)/\tau_R - 1/\tau_A \pm \sqrt{\{(-1+w_R)/\tau_R + 1/\tau_A\}^2 - 4k/(\tau_R\tau_A)}$, and no oscillation before learning requires no complex eigenvalues, that is, $\{(-1+w_R)/\tau_R + 1/\tau_A\}^2 - 4k/(\tau_R\tau_A) \geq 0$ (solid curve in *Figure 5—figure supplement 1*). Another condition is that the second peak is lower than the first peak in the successive presentation of two novel stimuli. To derive analytical expression, we made the following assumptions − i) neural activity changes linearly during the rising and decaying phases, and ii) during the rising phase, the adaptation variable $\bar{a}$ and external input are constant. We denote $t_0$ and $t_1$ as the duration of the rising and decaying phases, $r_0$ and $r_1$ are activities at the end of the rising and decaying phases with $\bar{r} = 0$ and $\bar{a} = 0$ as the baseline before the stimulus presentation. Also, if we denote $I_0$ as the constant input during the rising phase, then approximately, $r_0 = I_0 t_0/\tau_{eff}$ where $\tau_{eff} = \tau_R/(1 - w_R)$. During the decaying phase of the first stimulus presentation, $\bar{r}$ decreases linearly from $r_0$ to $r_1$, and then at the end of the presentation of the first stimulus, $\bar{a} = r_0(1 - exp(-t_1/\tau_A)) + (r_1 - r_0)\{1 - \tau_A/t_1(1 - exp(-t_1/\tau_A))\} \equiv a_1$.

Now based on the second assumption during the rising phase of the second stimulus presentation, the input becomes $I_0 - a_1 k$ and the expression for the second peak becomes $(I_0 - a_1 k)\frac{t_0}{\tau_R/(1-w_R)} + r_1$. Then the condition that the second peak is lower than the first peak gives $(I_0 - a_1 k)\frac{t_0}{\tau_R/(1-w_R)} + r_1 < r_0$. Replacing $I_0$ using $r_0 = I_0 t_0/\tau_{eff}$ leads to the condition $\frac{r_1/r_0 \cdot \tau_R/t_0}{1 - exp(-t_1/\tau_A) - (1 - r_1/r_0)(\tau_A/t_0(exp(-t_1/\tau_A) - 1) + 1)} < (1 - w_R)k$ (dotted curve in *Figure 5—figure supplement 1*).

## Network simulation in *Figure 4*

In *Figure 4*, we illustrated the dynamics of an example network with synaptic plasticity in feedforward and recurrent connections, and spike adaptation mechanisms. The network dynamics follows *Equation (3)* and as in the mean-field dynamics, we assumed linear dynamics with $\Phi(x) = x$ and uniform recurrent connectivity before learning $W_{ij}^R = w_R/N$. The input was modeled as a sum of a constant input $I_{const}$ and time-varying one which is the sum of two exponential functions $I_{dyn} = exp(-t/t_1) - exp(-t/t_2)$ with its strength $\xi_i$ varying across neurons as $I_i^X = I_{const} + \xi_i I_{dyn}$. For recurrent synaptic plasticity, Hebbian learning rule such as $\Delta W_{ij}^R = \frac{\alpha}{Nvar(\xi)}\xi_i(\xi_j - \bar{\xi})$ was considered where $\alpha$ is the strength of the plasticity. For feedforward synaptic plasticity, uniform scaling down of the time-varying input was considered such that $I_i^X$ changes to $I_i^X = I_{const} + \gamma\xi_i I_{dyn}$ after learning.

For the successive presentation of two stimuli, the changes in the recurrent connection become $W_{ij}^R \rightarrow W_{ij}^R + \Delta W_{ij}^{R,1} + \Delta W_{ij}^{R,2}$ with uncorrelated patterns $\xi_i^1$ and $\xi_i^2$. The duration of each stimulus presentation is denoted as $P_1$ and the external input correlated with one stimulus decays linearly during $P_2$ when another stimulus is on. The parameters used in *Figure 4* are $N = 2000$, $w_R = 0$, $k = 1.8$, $\tau_R = 5$ ms, $\tau_A = 200$ ms, $t_1 = 150$ ms, $t_2 = 50$ ms, $P_1 = 150$ ms and $P_2 = 100$ ms. $\xi_i$ is assumed to follow a gamma distribution with shape parameter three and $\alpha = 0.9$. $I_{const}$ is adjusted so that the baseline firing rate is 5 Hz, and $\gamma = 0.4$.

## Fitting experimental data in *Figures 5,6*

In *Figure 5*, activities in ITC neurons for the dimming detection task were fitted using the mean field dynamics given in *Equation (4)*. Under the assumption of a homogeneous network, activities to different stimuli can serve as a surrogate for activities of different neurons to one stimulus. Thus, we took an average of firing rates over stimuli (eight novel stimuli and 10 familiar stimuli for each neuron) and over 41 neurons classified as putative excitatory neurons (see more details in *Lim et al.,*

*2015*). Note that as the time-course data without taking an average over neurons is noisy and the number of stimuli is small, only the parameters for the mean-field dynamics could be inferred from the data from the dimming-detection task.

Before learning, the response dynamics is only determined by average variables $\bar{r}$ and $\bar{a}$, and by the parameters $w_R$, $k$, $\tau_R$, and $\tau_A$, which need to reproduce the data showing no damped oscillations before learning and suppressed response to the presentation of the second novel stimuli (*Figure 5—figure supplement 1*). Given $w_R$, $k$, $\tau_R$, and $\tau_A$, the external input was modeled as the sum of two exponential functions $a\exp(-t/t_1) - b\exp(-t/t_2) + b - a$ and their parameters are chosen such that the simulation fits the activities before learning (*Figure 5A*). Note that since we assume that the inhibitory activities follow the excitatory activities instantaneously, $w_R$ represents $W^{EE} - W^{EI}W^{IE}$, and can be negative. In the dynamics of $m$ and $n$ after learning, the strength of positive feedback is $\overline{fg_R}$ in *Equation (4)*, analogous to potentiation in high firing rate neurons. Together with parameters for slow adaptation currents, $\overline{fg_R}$ should be chosen to generate the oscillation with the period around 150 ms (*Figure 5—figure supplement 2A,B*). The external input for $m$ is also modeled as the sum of two exponential functions, and given $m$ and $n$ dynamics and the external input from the novel response, $\bar{f}_{R,F}$ were chosen to fit the magnitude of oscillation and reduction in firing rates in the mean response for familiar stimuli given the dynamics of $m$ (*Figure 5B*; *Figure 5—figure supplement 2C*).

In the simulation of the successive stimulus presentation in *Figure 5C D*, all the parameters are the same as in *Figure 5A B* and the external inputs for the first and second stimuli have the same temporal profile except for different onsets. During the presentation of the second stimuli, the external input for the first stimulus decays exponentially with a time constant of 50 ms. The parameters used in *Figure 5* are $w_R = 0$, $k = 1.8$, $\tau_R = 5$ ms, $\tau_A = 200$ ms, $\overline{fg_R} = 0.9$, $\bar{f}_R = 0.3$, $\bar{f}_F = -0.7$, $a = 6$, $b = 5$, $t_0 = 700$ ms, $t_1 = 40$ ms for the mean external input and $a = b = 5$, $t_0 = 400$ ms, $t_1 = 20$ ms for the external input of the $m$ dynamics. Note that for a wide range of $w_R$ with the same parameters except $a$, $b$, $t_0$, and $t_1$ adjusted to reproduce response to novel stimuli, the simulation fit the data well (*Figure 5—figure supplement 3*).

In the passive viewing task in *Figure 6*, responses to 125 novel and 125 familiar stimuli were measured, and 14 putative excitatory neurons were classified to show both potentiation and depression when the distributions of time-averaged activities before and after learning were compared (see more details in *Lim et al., 2015*). Time-course data at each rank of the stimuli in each neuron was noisy, and averaging over neurons was required to reduce noise. For this, we normalized activities in each neuron and took the average of these normalized activities over neurons at each rank (*Figure 1*).

To infer post-synaptic dependence of synaptic plasticity rules on normalized activities, we considered a network consisting of 125 neurons whose dynamics are described by *Equation (3)* and fit time-course data before and after learning. We set the parameters to be the same as in *Figure 5*, and fitted external inputs $I^X(t)$ and post-synaptic dependence of the feedforward and recurrent connections, $f_{R,F}$. The external input to each neuron was obtained to reproduce the response for novel stimuli at each rank as follows - Discretization of the dynamic equations in *Equation (3)* yields $\frac{\tau_r}{dt}\left(r_i^{nov}(t+dt) - r_i^{nov}(t)\right) = -r_i^{nov}(t) + \Phi\left(w_r\bar{r}_i^{nov}(t) - ka_i^{nov}(t) + I_i^X(t)\right)$ where $r_i^{nov}(t)$ is the firing rate for the novel stimulus at rank $i$, and $a_i^{nov}(t)$ is a low-pass filtered $r_i^{nov}(t)$. Given $w_R = 0$, $\tau_R = 5$ ms with the time step 5 ms to be the same as that in the data, the external input can be expressed as $I_i^X(t) = \Phi^{-1}\left(r_i^{nov}(t+dt)\right) + ka_i^{nov}(t)$, and thus, it is determined by activities for novel stimuli.

The post-synaptic dependence of the synaptic plasticity was obtained to fit the activities for familiar stimuli. As the single cell recordings do not allow inference on the pre-synaptic dependence, we assumed its form which is $g_{R,F} = 1$ for the highest rank and 0 otherwise such that $m$ in *Equation (4)* is the response to the familiar stimulus at the highest rank. In this case, discretization of the dynamic equation for familiar stimuli becomes $\frac{\tau_r}{dt}\left(r_i^{fam}(t+dt) - r_i^{fam}(t)\right) = -r_i^{fam}(t) + \Phi\left(w_r\bar{r}_i^{fam}(t) + f_{R,i}r_{max}^{fam}(t) - ka_i^{fam}(t) + I_i^X(t) + f_{F,i}I_{max}^X(t)\right)$. $f_{R,F}$ were fitted to mimic the response to familiar stimuli at each rank - the number of unknowns is 125 times 2 (125$f_R$ and 125$f_F$) and the number of data points to fit is 125 times 44 where 44 is the number of time steps so it is analogous to underdetermined system. We used the least square method with larger weights in the late phase to capture the rebound better (weight 5 from 230 ms after the

stimulus onset, and otherwise 1; different weights do not affect the performance qualitatively, not shown here). Note that in fitting and simulating the response to familiar stimuli, we used $r_{max}^{fam}(t)$ from the data to prevent the fitting error in $r_{max}^{fam}(t)$ from spreading over the network.

For nonlinear dynamics in *Figure 6—figure supplement 1*, the transfer function $\Phi(x)$ was obtained from the time-averaged response for novel stimuli – for each rank of novel stimuli, we took the time-averaged response in the time window between 75 ms and 200 ms after stimulus onset. Under the assumption that the transfer function $\Phi(x)$ is monotonically increasing and the distribution of synaptic inputs to novel stimuli follow Gaussian statistics, the transfer function is obtained by matching the input current and time-averaged response at the same rank (*Lim et al., 2015*).

## Models for alternative negative feedback mechanisms

Replacing adaptation $a_i$ in *Equation (3)* as $a^I$ which is an exponential filtered $\bar{r}$ with strength $k^I$ and time constant $\tau_A$, we can derive the mean-field dynamics of the model for the global inhibition as

$$\tau_R \frac{d\bar{r}}{dt} = -\bar{r} + w_R \bar{r} + \bar{f}_R m - k^I \bar{a}^I + \bar{I}_X + \bar{f}_F I_{FX}$$
$$\tau_A \frac{d\bar{a}^I}{dt} = -\bar{a}^I + \bar{r}$$
$$\tau_R \frac{dm}{dt} = -m + \overline{fg}_R m + I_{MX} + \overline{fg}_F I_{FX}$$

(6)

which is similar to *Equation (4)*, but without $n$ dynamics.

The short-term depression is modeled by a variable $x$ which represents the fraction of resources available after the depletion of neurotransmitters and therefore adjusts the strength of the synaptic connections (*Tsodyks and Markram, 1997*). The network activity is thus described by the following equations

$$\tau_r \frac{dr_i}{dt} = -r_i + \sum W_{ij}^R x_j r_j + I_j^X$$
$$\frac{dx_j}{dt} = \frac{1-x_j}{\tau_x} - \gamma x_j r_j$$

.

where $\tau_X$ and $\gamma$ represent the time constant and strength of short-term depression. Before the stimulus presentation, $x$ is initialized to its steady states given the parameters and baseline activity, and $W^R$ before and after learning is the same as in *Equation (3)*. To see whether short-term depression can reproduce the oscillatory response after learning, we considered the case that the recurrent connection is weak before learning, and the oscillation in the network is led by that in the high rate neurons. With larger pre-synaptic dependence $g_R$ for the high rate neurons, their dynamics can be approximated as

$$\tau_r \frac{dr_m}{dt} = -r_m + f_m^R g_m^R x_m r_m + I_m^X$$
$$\frac{dx_m}{dt} = \frac{1-x_m}{\tau_x} - \gamma x_m r_m$$

.

We fitted the parameters $f_m^R g_m^R$ and $\gamma$ analogous to the strengths of long-term synaptic plasticity and short-term plasticity, respectively (*Figure 6—figure supplement 2*). When we set $\tau_R$ = 5 ms, $\tau_x$ = 200 ms, $I_m^X$ was obtained from the maximal response to the novel stimuli. The best fitting parameters to the maximal response to the familiar stimuli are $f_m^R g_m^R$ = 2.56 and $\gamma$ = 0.125, and the time course with the best-fitted parameters cannot generate oscillation (*Figure 6—figure supplement 2*).

## Models for competitive interactions between two stimuli

Experimentally, stronger oscillation at around 5 Hz was observed in the presence of another stimulus in the visual field, which was accounted for by competitive interactions between the populations selective to each stimulus (*Moldakarimov et al., 2005*; *Rollenhagen and Olson, 2005*). Following these previous works, we considered two mutually inhibitory populations where each population is selective to one of two stimuli and its dynamics follow the dynamics of $m$ in the mean-field description under the presence of a single stimulus. Then the dynamics of two populations are given as follows:

$$\tau_R \frac{dm_i}{dt} = -m_i + \Phi\left(\overline{fg}_R m_i - kn_i - w_m j + I_{m,i}\right)$$
$$\tau_A \frac{dn_i}{dt} = -n_i + m_i$$

where population indices $i,j$ = 1 or two where $i \neq j$, and $w_c$ denotes the strength of the mutual

inhibition, set to be 0.1. $\Phi$ is the input current-output rate transfer function which was assumed to be piece-wise linear as $\Phi(x) = x$ for $x \geq -3$ and 0 otherwise. The remaining parameters and variables are the same as in **Figure 5** as $\overline{fg_R} = 0.9$, $k = 1.8$, $\tau_R = 5$ ms, $\tau_A = 200$ ms, $I_{m,i} = exp(-t/t_1) - exp(-t/t_2)$ where $t_1 = 400$ ms and $t_2 = 20$ ms.

## Acknowledgements

The author is grateful to Nicolas Brunel for valuable feedback on the manuscript and discussions with the initial suggestion of the project, David Sheinberg and David Freedman for sharing their data, Loreen Hertäg and Panagiota Theodoni for valuable feedback on the manuscript, John Rinzel for valuable discussion, and anonymous reviewers whose comments helped to improve the manuscript significantly. This research was supported by Research Fund for International Young Scientists at National Natural Science Foundation of China, 31650110468, and the author acknowledges the support of the NYU-ECNU Institute of Brain and Cognitive Science at NYU Shanghai.

## Additional information

### Funding

| Funder | Grant reference number | Author |
| --- | --- | --- |
| National Natural Science Foundation of China | Fund for International Young Scientists – 31650110468 | Sukbin Lim |

The funders had no role in study design, data collection and interpretation, or the decision to submit the work for publication.

### Author contributions

Sukbin Lim, Conceptualization, Data curation, Software, Formal analysis, Funding acquisition, Validation, Investigation, Visualization, Methodology, Writing—original draft, Writing—review and editing

### Author ORCIDs

Sukbin Lim  https://orcid.org/0000-0001-9936-5293

### Decision letter and Author response

Decision letter https://doi.org/10.7554/eLife.44098.026
Author response https://doi.org/10.7554/eLife.44098.027

## Additional files

### Supplementary files

• Source code 1. Data for *Figures 1* and *6*.
DOI: https://doi.org/10.7554/eLife.44098.018
• Transparent reporting form
DOI: https://doi.org/10.7554/eLife.44098.024

### Data availability

All data and codes used in the manuscript have been provided as source code files. Please note that the original data were generated in Sheinberg's and Freedman's labs, and the uploaded data are processed data used for network simulations and fitting.

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
