## [Decision Letter]

Thank you for submitting your article "Mechanisms underlying sharpening of visual response dynamics with familiarity" for consideration by *eLife*. Your article has been reviewed by two peer reviewers, one of whom is a member of our Board of Reviewing Editors, and the evaluation has been overseen by Michael Frank as the Senior Editor. The reviewers have opted to remain anonymous.

The reviewers have discussed the reviews with one another and the Reviewing Editor has drafted this decision to help you prepare a revised submission.

Summary:

In this paper, Lim addresses questions related to the cortical mechanisms that support learning through an innovative modeling-based approach. The specific focus of this paper is a model of how the stimulus-evoked response dynamics of IT neurons along timescales of a few hundred ms differ for novel as compared to highly familiar images. Previous experimental work found that familiarity leads to a sharp reduction in firing rate following the initial peak (phasic) response, then a rebound of firing that when taken together resemble a damped oscillation. Modeling this response could provide important insights into learning mechanisms of the brain.

The author builds on her earlier model of IT familiarity acquisition, which uses changes in tuning functions to argue that familiarity plasticity resides in synaptic plasticity of recurrent connections within IT. Here she proposes 1) synaptic plasticity in the recurrent connections, 2) rate adaptation and 3) plasticity in the feedforward inputs, are sufficient to account for experimentally observed changes in visual response between novel and familiar images. In particular, this work extends the work Lim et al., 2015. It focuses on reproducing a damped oscillatory component of the visual response after learning, which was not present before learning in experimental data.

The reviewers find the work interesting and exciting, but have also identified a number of issues that must be addressed for the manuscript to be suitable for publication. They also have provided a number of suggestions for improvement.

Essential revisions:

1) The BCM type rule was derived assuming there is only plasticity in the recurrent network (Lim et al., 2015). Then it is used here with plasticity in the feedforward network. That seems inconsistent. If I understand correctly, the rule should be re-inferred with plasticity both in feedforward and recurrent connections in the first place.

2) Describing this rebound/oscillatory component mechanistically is interesting, in terms of fitting the data. However, the functional implications are less clear. The work could be extended to show the functional implications of a network with the 3 ingredients: plasticity in the feedforward, in the recurrent and the rate adaptation, leading to this transition between overshoot and damped oscillations.

3) Is this model fully consistent with the experimental results that motivate it? Specifically, wouldn't turning off the image still induce an oscillatory response due to the positive/negative recurrent feedback? This seems counter to work the author cites (Meyer et al., 2014 Figure 5C, D – notice how when a familiar image is presented then left with a blank screen 'F-' there is no oscillation).

4) The paper focuses exclusively on excitatory neurons – is there a good reason for this? Can the model account for both excitatory and inhibitory response dynamics?

5) The author needs to include all the information to reproduce the paper.

6) The writing should be improved for both technical expects as well as a general audience, particularly in the subsection “Interactions between synaptic plasticity and slow negative feedback”.

7) The author must provide code for the reviewers (both for the network with fitted parameters and the code for the fitting procedure) and post the code publicly after publication.

[Editors' note: further revisions were requested prior to acceptance, as described below.]

Thank you for submitting your article "Mechanisms underlying sharpening of visual response dynamics with familiarity" for consideration by *eLife*. Your article has been reviewed by two peer reviewers, one of whom is a member of our Board of Reviewing Editors, and the evaluation has been overseen by Michael Frank as the Senior Editor.

The reviewers have discussed the reviews with one another and the Reviewing Editor has drafted this decision to help you prepare a revised submission.

Summary:

The reviewers would like to thank the authors for all the work they did. The paper has improved as a result of these clarifications. A few remaining items should be addressed before the manuscript is accepted for publication.

Essential revisions:

1) In the first round of reviews, the reviewers pointed out that the BCM type rule was derived in 2015 assuming that plasticity was only in the recurrent network but in the current manuscript it was applied to a network with feedforward plasticity; the reviewers asked whether the rule should be re-derived. The author reply focused on a normalization procedure applied to the neural data. It is not clear to the reviewers how this normalization procedure relates to the issues raised about recurrent versus feedforward processing. Please justify the use of the BCM type rule in the model with multiple types of plasticity.

2) In the first round of reviews, the reviewers requested that the work be extended to show the functional implications of the proposed network. The authors responded by incorporating a paragraph into the Discussion highlighting the bridge between two classes of models, and these are nice points to make. However, the reviewers would like to clarify their request to complement the current presentation, which focuses on the network architectures required to recapitulate an experimentally-observed phenomenology, with more insight into the functional implications of this work. For example, beyond constraining mechanistic models, why do we care about the damped oscillatory response?

What are its functional implications for representation in IT and/or behavior? What types of functions is a network with the 3 proposed ingredients capable of?

---

## [Author Response]

Essential revisions:1) The BCM type rule was derived assuming there is only plasticity in the recurrent network (Lim et al., 2015). Then it is used here with plasticity in the feedforward network. That seems inconsistent. If I understand correctly, the rule should be re-inferred with plasticity both in feedforward and recurrent connections in the first place.

Thank you for pointing out the lack of clarity concerning connection to the previous work (Lim et al., 2015). As noted by the reviewers, one of the main findings in my previous work (Lim et al., 2015) was that when the synaptic plasticity was inferred from changes of time averaged response with learning, the post-synaptic firing rate separating depression and potentiation, denoted as a threshold θ, was strongly correlated with mean and standard deviation of post-synaptic firing rates. This is reminiscent of the BCM rule claiming the dynamic evolution of θ depending on the post-synaptic activities, although our work compared snapshots of statistics before and after learning, and examined θ across different neurons, that is, the spatial version of the BCM rule. Such a correlation between θ and post-synaptic activities may arise from similar synaptic plasticity rule across different neurons when input currents and firing rates are normalized. Author response image 1, adapted from the previous paper (Figure 4 in Lim et al., 2015) shows that when firing rates and inputs were normalized by the mean and standard deviation of them before learning, such a correlation disappeared (g and h compared to e and f). This may suggest the same synaptic plasticity rule across different neurons when post-synaptic rates were normalized (c), and as overall firing rates changes across different neurons, the threshold changes proportionally as θ = σθ’+μ where θ’ is the threshold for normalized firing rate, and μ and σ are the mean and standard deviation of activities.

Inspired by this previous observation, we inferred synaptic plasticity from normalized activities when it was allowed – in the passive viewing task in Figures 1 and 6, we normalized activities in each neuron using the mean and standard deviation of activities before learning, and averaged such normalized activities over different neurons. Note that averaging over neurons was required to reduce noise in the time-course data, in contrast to the previous work comparing the time-averaged responses and deriving recurrent synaptic plasticity rule in individual neurons. Under the assumption that synaptic plasticity rule is the same across different neurons when inputs and rates are normalized, the correlation between the threshold and post-synaptic activities naturally arises even if depression occurs mainly in the feedforward connections and potentiation occurs in the recurrent connections.

We now add a sentence in Results and the paragraph in Discussion to clarify this as follows:

Results: “… Note that the synaptic plasticity was inferred from normalized firing rates averaged over neurons under the assumption that the dependence of synaptic plasticity rules on normalized firing rates is the same across different neurons (See Discussion for justification).”

Discussion: “Here, we extended our previous work inferring recurrent synaptic plasticity rules from time-averaged data in a static model of a cortical network to time-course data and a dynamic model with additional spike adaptation mechanisms and feedforward synaptic plasticity (Lim et al., 2015). […] In this case, the dependence of synaptic plasticity rules on post-synaptic rates scales proportionally to changes of a range of firing rates, resulting in a correlation between the threshold and neural activity even with synaptic plasticity in different connections, consistent with the previous observation from time-averaged responses (Lim et al., 2015).”

2) Describing this rebound/oscillatory component mechanistically is interesting, in terms of fitting the data. However, the functional implications are less clear. The work could be extended to show the functional implications of a network with the 3 ingredients: plasticity in the feedforward, in the recurrent and the rate adaptation, leading to this transition between overshoot and damped oscillations.

We thank the reviewers for pointing out the lack of clarity. The current work proposed the conditions on synaptic plasticity in recurrent and feedforward connections, and spike-adaptation mechanisms for reproducing the time course activities. Recurrent synaptic plasticity broadens the distribution of activities as typical Hebbian synaptic plasticity, while depression in feedforward inputs decreases average firing activities. Thus, synaptic plasticity enables the sparse and efficient representation of the learned stimuli. On the other hand, adaptation-like mechanisms are critical for shaping the dynamics as the network cannot generate oscillatory response without slow negative feedback (Figure 2). It has been suggested that stronger oscillatory response after learning might be important for putting neurons to be ready for the following stimuli as suppressing activity strongly in particular in the late phase of the stimulus presentation. Thus, the slow adaptation mechanisms together with synaptic plasticity may play an important role in the rapid processing of the learned stimuli.

The functional role of these three main ingredients have been investigated previously in relation to different types of recognition memory. For familiarity detection, different forms of feedforward synaptic plasticity have been explored to reproduce a lower response for familiar stimuli, but without considering response dynamics (Bogacz and Brown, 2003; Norman and O'Reilly, 2003; Sohal and Hasselmo, 2000). On the other hand, most theoretical works implementing synaptic plasticity in recurrent connections have focused on associative memory and the emergence of attractors with learning (Amit and Brunel, 1997; Pereira and Brunel, 2018; Sohal and Hasselmo, 2000). Finally, the adaptation mechanisms in the temporal cortex have been suggested to encode the recency of stimuli, which is typically measured by suppression of the response to the repetition of a stimulus (Meyer and Rust, 2018; Miller, Li and Desimone, 1991; Vogels, 2016; Xiang and Brown, 1998). As the time scale of repetition suppression lasts up to seconds, the spike adaptation mechanism considered in the current study may only encode the recency signal on a shorter time scale.

Most of these works focused on one of the three ingredients, except Sohal and Hasselmo, 2000, whose model contained feedforward and recurrent synaptic plasticity as well as spike adaptation mechanisms on a longer time scale. Still, as other works, Sohal and Hasselmo proposed each ingredient for a different type of memory, and did not investigate their interactions. On the other hand, we focused on familiarity detection and investigated how the three ingredients shape response dynamics together. Recently, Pereira and Brunel, 2018, have investigated the capacity of associative memory with recurrent synaptic plasticity whose form was derived from neural activities related to familiarity detection. Similarly, it can be further investigated how the three ingredients derived in the current study contribute to other types of memory like associative memory and recency effect, and how a memory capacity for familiarity detection changes dynamically during the stimulus presentation with the spike adaptation mechanisms. However, this is beyond the scope of the current paper.

The functional implications of each mechanism are summarized in the first paragraph in Discussion and to discuss them related to the previous works, we modified the paragraph in Discussion as follows:

“Our work provides a reconciling perspective between two prominent classes of synaptic plasticity models suggested for familiarity detection and associative memory in ITC. […] Similarly, it can be further investigated how the feedforward and recurrent synaptic plasticity rules derived from the data for familiarity detection contribute to other types of memory, and how a memory capacity changes dynamically during the stimulus presentation with slow spike adaptation mechanisms.”

3) Is this model fully consistent with the experimental results that motivate it? Specifically, wouldn't turning off the image still induce an oscillatory response due to the positive/negative recurrent feedback? This seems counter to work the author cites (Meyer et al., 2014 Figure 5C, D – notice how when a familiar image is presented then left with a blank screen 'F-' there is no oscillation).

To reveal the mechanisms underlying changes in response dynamics with learning, we mainly considered the experimental results obtained from three different laboratories. We note that different experimental settings may lead to quantitatively different results even if the underlying principle is the same.

The data obtained from a passive viewing task with a larger number of stimuli suggests that oscillation is originated from the response dynamics of excitatory neurons to their most preferred stimuli about the top 5% of the stimuli (Figure 1C). As different data sets from the dimming-detection task or with the successive presentation of two stimuli (Meyer et al., 2014) employed the smaller number of stimuli (10 familiar stimuli for each neuron in the dimming detection task, and on average 6.5 times 2 familiar stimuli for Meyer et al., 2014) in comparison to 125 stimuli in the passive viewing task, it requires sampling at least ten times more neurons to see the oscillation. Furthermore, as addressed in the reply to point 4 below, damped oscillation is not prominent in inhibitory neurons. Thus, averaging response dynamics over excitatory neurons and inhibitory neurons can mask oscillation.

The response to a short presentation of familiar stimuli shown as blue curves in Figure 5C, E, and G (adapted from Meyer et al., 2014) is less consistent as panels C and E do not show oscillation while panel G shows oscillation. Such inconsistency may arise from sparse sampling from highly oscillatory excitatory neurons and averaging the response over both excitatory and inhibitory neurons. Similarly, in Figure 5D compared to Meyer et al., 2014, the network with the parameters inferred from one experimental setting might be difficult to reproduce the data from another experiment quantitatively. Instead, we focused on inferring the mechanisms underlying visual learning from the common qualitative features across the different experiments showing stronger oscillation after learning although their magnitude may depend on different sampling of neurons and stimuli.

To clarify this, we added the following sentence in “Effects of visual learning on response dynamics” section in Results:

“… Note that although all three experiments suggest stronger oscillation after learning, its strength may vary depending on a sampling of neurons and stimuli as only excitatory neurons with their most preferred stimuli exhibit strong oscillation after learning (Figure 1D).”

4) The paper focuses exclusively on excitatory neurons – is there a good reason for this? Can the model account for both excitatory and inhibitory response dynamics?

We thank the reviewers for bringing up the lack of clarity in simplification on the inhibitory dynamics in the model and the possible role of inhibitory dynamics and synaptic plasticity in shaping response dynamics. As the reviewers noted, the dynamics of inhibitory neurons follow that of mean excitatory neurons in the model, and the recurrent inputs to excitatory neurons reflect the feedback through inhibitory neurons as the connectivity strength *w_R_* represents *W^EE^*_−_
*W^EI^ W^IE^,*and can be negative. Such a simplification is permitted under the assumption that inhibitory dynamics do not provide a major contribution to the emergence of oscillation after learning, which can be justified for the following reasons. First, no dependence of input changes on post-synaptic firing rates in inhibitory neurons observed experimentally (Author response image 1) suggests that changes in inhibitory activities with learning can reflect the reduction of average excitatory activities and thereafter, excitatory inputs to inhibitory neurons without synaptic plasticity in the excitatory (*E*)-to-inhibitory (*I*) connections (Lim et al., 2015). On the other hand, anti-Hebbian synaptic plasticity in the *I*-to-*E* connections can have similar effects as Hebbian-synaptic plasticity in the *E*-to-*E* connections. Alternatively, overall potentiation in the *I*-to-*E* connections can provide stronger negative feedback or can replace the role of feedforward synaptic plasticity. However, as the dynamics of inhibitory neurons show strong suppression almost to the baseline in the late phase of the stimulus presentation after learning (150-200 ms after the stimulus onset in Figure 1—figure supplement 1), neither anti Hebbian synaptic plasticity nor potentiation can account for an increase of maximal response of excitatory neurons in the early phase or overall reduction in activities in the late phase (Figure 1). Thus, although inhibitory dynamics or synaptic plasticity can be complementary to the mechanisms addressed in the paper, we think that it cannot be a major factor to lead to oscillation with stronger positive or negative feedback, or reduction in activities with learning.

Another reason why we did not model the inhibitory dynamics explicitly is difficulty in fitting the data quantitatively – in the passive viewing task, as many stimuli were presented with a short inter-stimulus interval (200 ms for the stimulus presentation and 50 ms for the interval between the stimuli). Thus, the baseline activities before the stimulus presentation may reflect the residual activities in response to the previous stimulus. Indeed, inhibitory neurons show a more prominent effect having the baseline around 30 Hz which is too high compared to 10 Hz in Meyer et al. or other experiments. Such distortion in the temporal profiles hinders fitting the time course of inhibitory neurons quantitatively.

In summary, we made a simplification of the dynamics of inhibitory neurons based on the assumption that changes in inhibitory activities with learning can be explained by that of excitatory neurons, and inhibitory dynamics or synaptic plasticity does not provide a major contribution to the emergence of the oscillation. Also, the limitation of the current experimental settings poses difficulty in fitting the inhibitory dynamics quantitatively. To clarify this, we modified the text in Results about modeling excitatory dynamics only, the candidate mechanisms for reduction of average activities with learning and a paragraph in Discussion for the role of inhibitory dynamics or synaptic plasticity as follows:

Results: “In sum, the prominent effects of visual learning on responses of excitatory neurons are i) reduction in average response, ii) increase in maximum response, and iii) stronger oscillations after learning. […] Such a simplification is based on the experimental observation that input changes and the magnitude of rebound activity depend weakly on the postsynaptic firing rates in inhibitory neurons (see Discussion for further justification).”

Results: “Instead, reduction in average response requires changes in external inputs or other recurrent inputs such as suppression in other excitatory inputs or enhanced inhibition. […] This suggests that the effect of potentiated inhibition in the late phase is weaker than in the early phase while reduction of excitatory activities was observed in the late phase (Figure 1A, B).”

Discussion: “In our work, we assumed that inhibition minimally contributes to shaping response dynamics with learning for the following reasons. […] Thus, we assumed that changes in the inhibitory pathway are less likely to induce oscillation or suppression in the excitatory neurons…”

5) The author needs to include all the information to reproduce the paper.

We modified the Materials and methods section significantly and published the codes and data (see the reply to point 7).

6) The writing should be improved for both technical expects as well as a general audience, particularly in the subsection “Interactions between synaptic plasticity and slow negative feedback”.

We highly appreciate the reviewers’ constructive suggestions to improve the clarity in the paper. Following the first three suggestions, we modified the relevant texts in “Interactions between recurrent synaptic plasticity and slow negative feedback” section and created a new section, “Additional synaptic plasticity for a reduction in average response” in Results.

With regard to the last suggestion, we added a new paragraph in Discussion as addressing the relation to the BCM-rule (see the relevant paragraph in point 1).

7) The author must provide code for the reviewers (both for the network with fitted parameters and the code for the fitting procedure) and post the code publicly after publication.

In the original submission, I uploaded the code and data on Github and provided this information in Data availability. However, I did not include this information in the manuscript due to my carelessness, and now I added this information in Materials and methods as follows: “Code availability. The data analysis and network simulations were performed in MATLAB. The data, codes for fitting the data and network simulations are available at https://github.com/slim-compneuro/Dynamics_NovelvsFamiliar.”

[Editors' note: further revisions were requested prior to acceptance, as described below.]

Essential revisions:1) In the first round of reviews, the reviewers pointed out that the BCM type rule was derived in 2015 assuming that plasticity was only in the recurrent network but in the current manuscript it was applied to a network with feedforward plasticity; the reviewers asked whether the rule should be re-derived. The author reply focused on a normalization procedure applied to the neural data. It is not clear to the reviewers how this normalization procedure relates to the issues raised about recurrent versus feedforward processing. Please justify the use of the BCM type rule in the model with multiple types of plasticity.

Thank you for pointing out the lack of clarity. In the previous response, I addressed that under the assumption of the same learning rule across different neurons for normalized firing rates and input currents, a correlation between activity and the threshold separating the depression and potentiation emerges as observed in the previous work based on time-averaged response. However, the contribution of different synaptic plasticity such as feedforward or recurrent connections cannot be dissected using time-averaged response, and changes in temporal dynamics such as synchronous oscillations emerging after learning were further required.

In the time course data, as the response at each rank of stimuli in individual neuron was noisy, inferring both feedforward and recurrent synaptic plasticity in each neuron was not feasible. Instead, we assumed the BCM-type rules in each synaptic plasticity, that is, the same learning rule across different neurons as a function of the normalized firing rate. To validate this assumption, it may require more trials for the same stimulus or a larger set of stimuli so that averaging over a larger number of trials or stimuli at a similar rank can reduce noise and inference of synaptic plasticity in individual neurons would be allowed. Such a direct re-inference is not feasible in the current data set, given that oscillation is prominent in the top 5% of preferred stimuli, that is, 5-6 stimuli out of 125 stimuli, and averaging over stimuli would diminish the dependence of the learning rule on the post-synaptic activity.

On the other hand, as pointed in Figure 6A and its surrounding text, the rebound strength of damped oscillation for learned stimuli was similar to the post-synaptic dependence of recurrent synaptic plasticity. Although such rebound strengths at each rank of stimuli vary over neurons, its standard error of the mean was relatively small compared to the mean (shaded area compared to the solid line in Figure 1D). This may suggest a similar recurrent synaptic plasticity rule across different neurons as a function of the rank of stimuli, or equivalently normalized firing rates (Figure 6—figure supplement 3). If the learning rule inferred from time-averaged activities is the combination of recurrent and feedforward synaptic plasticity, the same synaptic plasticity of this mixture and recurrent connections across different neurons (Figure 6—figure supplement 3A and B) would lead to the same synaptic plasticity of feedforward connections as well (Figure 6—figure supplement 3C). Thus, a BCM-type of synaptic plasticity rules observed in time-averaged response and relatively small noise in rebound strengths across different neurons might provide indirect support for similar synaptic plasticity over neurons for both feedforward and recurrent synaptic plasticity.

To clarify this point, I modified the text in the Discussion (second paragraph) and added a supplementary figure (Figure 6—figure supplement 3).

2) In the first round of reviews, the reviewers requested that the work be extended to show the functional implications of the proposed network. The authors responded by incorporating a paragraph into the Discussion highlighting the bridge between two classes of models, and these are nice points to make. However, the reviewers would like to clarify their request to complement the current presentation, which focuses on the network architectures required to recapitulate an experimentally-observed phenomenology, with more insight into the functional implications of this work. For example, beyond constraining mechanistic models, why do we care about the damped oscillatory response?What are its functional implications for representation in IT and/or behavior? What types of functions is a network with the 3 proposed ingredients capable of?

I appreciate the reviewers for raising this issue again – in this revision, I found that I missed literature discussing the prevalence of low-frequency oscillation in the brain and their functional roles (Buzsaki, 2011). The frequency of damped oscillation analyzed in this work is around 5Hz, which is in the range of theta oscillations. Closely related to the visual response in ITC investigated in this work, such a low-frequency oscillation has been discussed in visual search to characterize overt exploration or sampling behaviors such as saccadic or microsaccadic eye movements (Otero-Millan, Troncoso et al., 2008; Buzsaki, 2011). Similarly, it was suggested that covert shift of attention samples different stimuli rhythmically at a similar frequency range (Landau and Fries, 2012; Fiebelkorn, Saalmann et al., 2013; Dugue, Marque et al., 2015).

In line with these studies, a low-frequency oscillation at the theta range has been observed in several electrophysiological studies in ITC while monkeys passively viewed the images or performed attention tasks (Nakamura, Mikami et al., 1991; Nakamura, Mikami et al., 1992; Sheinberg and Logothetis, 1997; Freedman, Riesenhuber et al., 2006; Woloszyn and Sheinberg, 2012). In particular, Rollenhagen and Olson observed that low-frequency oscillation became stronger when an object eliciting an excitatory response was present together with a flanker stimulus that alone elicits little or no activity (Rollenhagen and Olson, 2005). The authors and Moldakarimov et al. in the following theoretical study suggested competitive interactions between populations representing each stimulus can generate such oscillation together with fatigue mechanisms (Moldakarimov, Rollenhagen et al., 2005; Rollenhagen and Olson, 2005). The frequency of enhanced oscillation was around 5 Hz, consistent with other electrophysiology and behavior studies, and thus, the oscillatory response may suggest perceptual alternation between different stimuli and covert shift of attention.

Interestingly, the frequency of damped oscillation in the presentation of a single familiar stimulus is similar to that observed during the visual search. Based on the adaptation mechanisms proposed in the current work which determines the frequency of oscillation, competition between two different familiar stimuli can generate stronger oscillation at a similar frequency. To show this, I utilized the mean-field dynamics considered in Figure 5, which generated damped oscillation in the presentation of a single familiar stimulus. Considering two mutually inhibitory populations each of which mimics the maximum response to a single familiar stimulus (Figure 7A), the experimental observation was reproduced qualitatively – the onset of the second stimulus transiently suppressed the response to the first stimulus and the oscillation during the presentation of both stimuli becomes stronger (Figure 7B). Furthermore, the oscillation frequency for the single stimulus presentation and presentation of two stimuli is similar around 5 Hz. This may indicate that low-frequency damped oscillators for a single familiar stimulus can be a building block for a rhythmic sampling of multiple stimuli and attentional shift through competitive interactions.

To discuss this possible functional role, I added a new paragraph in the Discussion (sixth paragraph), Figure 7, and a section describing the competition model in Materials and methods (subsection “Models for competitive interactions between two stimuli”).